# Novel Insights into Factor D Inhibition

**DOI:** 10.3390/ijms23137216

**Published:** 2022-06-29

**Authors:** Eleni Gavriilaki, Anna Papakonstantinou, Konstantinos A. Agrios

**Affiliations:** 1Hematology Department, G Papanicolaou Hospital, 57010 Thessaloniki, Greece; 2Department of Urology, Aristotle University of Thessaloniki, 54124 Thessaloniki, Greece; annapapak86@gmail.com; 3Department of Chemistry, Villanova University, 800 Lancaster Ave., Villanova, PA 19085, USA

**Keywords:** complement, factor D, paroxysmal nocturnal hemoglobinuria

## Abstract

Complement-mediated diseases or complementopathies, such as Paroxysmal nocturnal hemoglobinuria (PNH), cold agglutinin disease (CAD), and transplant-associated thrombotic microangiopathy (TA-TMA), demand advanced complement diagnostics and therapeutics be adopted in a vast field of medical specialties, such as hematology, transplantation, rheumatology, and nephrology. The miracle of complement inhibitors as “orphan drugs” has dramatically improved morbidity and mortality in patients with otherwise life-threatening complementopathies. Efficacy has been significantly improved by upstream inhibition in patients with PNH. Different molecules may exert diverse characteristics in vitro and in vivo. Further studies remain to show safety and efficacy of upstream inhibition in other complementopathies. In addition, cost and availability issues are major drawbacks of current treatments. Therefore, further developments are warranted to address the unmet clinical needs in the field of complementopathies. This state-of-the-art narrative review aims to delineate novel insights into factor D inhibition as a promising target for complementopathies.

## 1. Introduction

### 1.1. PNH as the Disease Model of Complementopathies

Complement-mediated diseases or complementopathies, such as paroxysmal nocturnal hemoglobinuria (PNH), atypical hemolytic uremic syndrome (aHUS), cold agglutinin disease (CAD), and transplant-associated thrombotic microangiopathy (TA-TMA), demand advanced complement diagnostics and therapeutics be adopted in a vast field of medical specialties, such as hematology, transplantation, rheumatology, and nephrology [1]. Complementopathies manifest through complex mechanisms, which impact disease manifestations. PNH is the prototype disease model as the first indication of complement inhibition. It is a rare, acquired hematological disease caused by somatic alterations in the PIGA gene [2], leading to the disease’s clinical manifestations, such as complement-mediated hemolysis and thrombosis [3].

In this context, research has focused on blocking complement pathway targets, to reduce complement-mediated effects, resulting in the introduction of C5 inhibitors, initially eculizumab [4], followed by ravulizumab [5]. Both have been approved for the treatment of PNH. Additional C5 inhibitors are under advanced clinical development [6]. Nonetheless, continuing extravascular hemolysis remains to be addressed through novel molecules, which aim at upstream inhibition in the complement pathway [7]. Patients’ C3 opsonized red blood cells are susceptible to destruction from active macrophages [8] while on anti-C5 treatment, resulting in extravascular hemolysis and introducing a surrogate method of erythrocyte lysis in PNH individuals [9].

The apparent scientific and commercial success of eculizumab (Soliris^R^) and C5 inhibition ignited the interest of other drug discovery laboratories. They have targeted additional complement components in order to develop novel therapeutics with improved clinical outcomes [10]. A C3 inhibitor known as pegcetacoplan, a subcutaneously administered, pegylated cyclic peptide, was recently approved as a novel PNH treatment [11]. Multiple other molecules are under advanced clinical development.

### 1.2. Other Complementopathies

In the field of TMAs, aHUS has been the disease model for complement inhibition, characterized by excessive complement activation in patient sera due to mutations in complement-related proteins or acquired autoantibodies. Both eculizumab and ravulizumab have been approved for aHUS treatment.

Beyond aHUS, both C3 glomerulonephritis (C3GN) and dense deposit disease (DDD) are classified as a set of progressive kidney diseases, known as C3 glomerulopathy (C3G). C3G manifests most often through complement dysregulation of the alternative pathway. However, terminal pathway dysregulation is another route for the pathogenesis of the disease [12]. Autoantibodies produced aim at the C3 or C5 convertase, increasing the half-life of these important, normally short-lived proteins. This process dysregulates the alternative pathway and induces C3 glomerular accumulation and end-stage renal disease [13]. This disease has a high recurrence risk in kidney transplant patients and is currently under advanced clinical trials of complement inhibition, including factor D inhibitors [14].

Age-related macular degeneration (AMD) is a neurodegenerative disease of the elderly and one of the three main causes of blindness [15]. Apart from age, genetic factors as well as environmental factors, such as nutrition, exposure to light, and smoking, contribute to the pathogenesis of the disease [16]. AMD has been associated with the alternative complement pathway, when the activated C3 starts a process of “autoactivation”, while interacting over decades with specific surfaces in the retina [16]. This process eventually leads to the irreversible loss of central vision. Complement inhibition in AMD has produced encouraging results, and research is ongoing with three phase 3 trials and four phase 2 trials [17]. More specifically, factor D (FD) antibody lampalizumab, intravitreally injected, has been shown to improve AMD in the MAHALO study, a phase 2 trial, thus paving the way towards two phase 3 trials [16], which did not validate this result, exhibiting the need for more research [17].

### 1.3. The Complement System

The complement system plays a vital role in innate immunity. Complement activation occurs via three distinct cascades termed the classical (CP), lectin (LP), and alternative (AP) pathways of complement (Figure 1) [18]. The key step in complement activation is the cleavage of a single peptide bond on the α-chain of the soluble third component of complement, C3, catalyzed by protein complexes known as C3 convertases. The CP and the LP are initiated by the binding of recognition proteins (PRPs) to specific targets. The CP is initiated by the binding of antigen–antibody complexes to the C1q protein. On the other hand, the LP does not recognize antibodies bound to their targets and is triggered with mannose-binding lectin (MBL) or ficolins and collectins binding to repeating carbohydrate moieties found on the surface of microbial pathogens. In contrast to the CP and the LP, the AP of complement is capable of auto-activation via a low-level steady process occurring in the fluid phase, known as C3 “tickover” (Figure 1). The AP can also be initiated as an “amplification loop” orchestrated by surface bound C3b that is generated by CP or LP activation. It has been shown that AP amplification is responsible for more than 80% of C5 cleavage when initial activation occurred via the CP [19].

Therefore, the continuous control of AP activation is absolutely critical for proper complement function and failure of this, for a variety of reasons, has been shown to be the cause of complementopathies [18]. Moreover, in vivo studies using gene-targeted mice have provided strong evidence that two serine proteases, namely Factor D (FD) and Factor B (FB), are essential for AP activation [20,21]. As a result, both FB and FD recently emerged as promising complement therapeutic targets that allowed the discovery of small molecule inhibitors, suitable for oral administration, promising to treat complementopathies [22,23]. AP inhibition, by targeting its unique components FB and FD, would stop excessive complement activation without shutting down the function of CP and LP. By means of mechanism, FD inhibition should prevent the formation of AP C3 convertase whereas FB inhibition should block its activity. It is important to note that directly targeting the C3 protein would stop C3 activation by all convertases and extend beyond AP inhibition. It is of note that protein fragments C3a and C5a, produced by the cleavage of C3 and C5, respectively, are pro-inflammatory molecules that can also exacerbate certain complement-mediated rare disorders by activating their respective protein receptors (Figure 1). Avacopan, an orally administered small molecule C5aR antagonist was recently approved for the treatment of ANCA-associated vasculitis [24]. Therefore, this state-of-the-art narrative review aims to delineate novel insights into factor D inhibition as a promising target for complementopathies.

## 2. Factor D Inhibition

### 2.1. Factor D

FD is an attractive target for pharmacological intervention because it has the lowest concentration of any complement protein in human blood (1.8 ± 0.4 µg/mL) making it the limiting enzyme in the activation sequence of the AP [25]. Factor D is categorized as an S1 type serine protease (24 KD, 228 amino acids), which is largely produced in adipocytes. It catalyzes the hydrolysis of a single Arg_234_-Lys_235_ (P1-P1′) bond in FB complexed with C3 (H_2_O) or C3b. Structurally, all serine proteases share four features essential for catalysis: (a) a catalytic triad, (b) an oxyanion binding hole, (c) a specificity binding pocket, (d) a non-specific binding site. With the exception of the oxyanion binding hole, the other three elements of the FD catalytic site have atypical conformations making FD an intriguing drug discovery target. FD circulates in blood in a mature but inactive, self-inhibited (latent) form or conformation [26,27]. Crystallographic studies on the native, as well as bound with covalent inhibitors, protein have provided very useful insights into the distinct conformational regulation of FD proteolytic activity [28,29,30]. The presence of a unique twisted salt bridge between the substrate recognizing amino acid Asp_189_ and Arg_218_ elevates surface loop 214–218 and generates a deep and narrow S1 specificity pocket [28]. The positioning of the surface loop 214–218, known as the self-inhibitory loop, renders the enzyme catalytically inactive by (a) preventing the P1 to P3 residues of the substrate from forming anti-parallel β-sheets with the non-specific substrate binding site and (b) not allowing the typical (canonical) alignment of the catalytic triad (Asp_102_-His_57_-Ser_195_). Specifically, Ser_215_, which is engaged in a hydrogen bonding interaction with Asp_102_, forces His_57_ to assume the *trans* (outward) conformation and to point away from Ser_195_ [28]. The elegant crystal structure of the complex C3bFBFD (S195A) has confirmed the proposed mechanism of FD activation and has shown how the membrane-bound C3 convertase (C3bBb) is formed following FD proteolytic action [29]. FD binds to an open FB conformation in its Mg^2+^-dependent complex with C3b, through a site distant to the catalytic center, where Arg_234_ and the scissile loop 224–239 of FB become exposed following a series of conformational changes. This binding event triggers reversible conformational changes in the FD catalytic center, including the rearrangement of the self-inhibitory loop. As a result, Arg_218_ points out of the S1 pocket and interacts with Factor B’s Glu_230_ (P5 residue) instead of being engaged in the previously mentioned salt bridge with Asp_189_. This is the active form or conformation of FD which allows the proper (canonical) topology of the catalytic triad [29,31]. The self-inhibitory mechanism as well as the substrate-induced activation make FD a self-regulating enzyme with very high substrate specificity and provide unique opportunities for therapeutic intervention within the AP of complement.

### 2.2. The Discovery of Small Molecule FD Inhibitors

**-** 
**Irreversible FD inhibitors**


Even though a lot of knowledge had been accumulated on the structural and functional biology of FD, medicinal chemistry efforts to identify small molecule FD inhibitors with therapeutic promise were not as successful. Isatoic anhydride [32] and 2,4-dichloroisocoumarin [33] are covalent, irreversible (known to be unattractive for drug development), and mechanism-based FD inhibitors (Figure 2). BCX1470 is another known irreversible FD inhibitor [34] carrying a very basic amidine moiety that is very likely recognized by the “open S1 conformation” of FD (active conformation) via a charged interaction with the Asp_189_ carboxylate prior to the covalent bond formation between the electrophilic ester and the nucleophilic Ser_195_. It is noteworthy that Nafamostat, a related benzamidine structure, exhibits some weak FD inhibition besides the inhibition of a protease panel in the coagulation and kallikrein–kinin systems [35].

#### 2.2.1. First-Generation of Latent FD Inhibitors

A series of small molecules owned by Novartis, built around an (*S*)-proline scaffold, appeared in the literature (2016) as the first potent and selective reversible latent FD inhibitors [23]. Moreover, they were shown to specifically block AP activation without any effect on the other two complement pathways. It is of note that (*S*)-proline-based peptidomimetic molecules were already known as serine–protease ligands. They bind to the S1 pocket and to the non-prime site (non-specific binding site) of typical serine proteases [36]. The Novartis team followed a rational structure-based drug design (SBDD) approach aided by complementary fragment-based screening (FBS), in silico docking, X-ray crystallography, and NMR spectroscopy experiments [37]. The use of computational algorithms pointed to the FD protein pockets S1, S1′ and S2′ as the active site hot spots. The calculated low ligandability scores for all three hot spots mandated the design of molecular structures carrying suitable functional groups capable of binding interactions with amino acid residues of all three pockets.

A moderate inhibitor (molecule **1**, Figure 3) of kallikrein 7 (KLK7), a related S1 serine protease, without any inhibitory activity against human FD in an enzymatic thioesterolysis assay [38], provided the blueprint for the rational design of latent FD inhibitors. The co-crystal structure of molecule **1** with KLK7 revealed the binding motif pictured in schematic structure **2**, where the (S)-configured proline core occupied the S1′ pocket, while the western and eastern aryl moieties were oriented towards the S1 and S2′ pockets, respectively. The design of latent FD inhibitors based on structural motif **2** entailed in silico modeling followed by chemical synthesis of the resulting hits and led to the first reversible, albeit weak FD inhibitor, molecule **3A** (IC_50_ = 14 μM) that also blocked KLK7 activity (IC_50_ = 9 μM). Two essential H-bond interactions, one between the urea carbonyl oxygen and the Gly_193_ -NH- (near the oxyanion hole) and the other between the eastern amide -NH- and the backbone carbonyl oxygen of Leu_40_, act as the anchors of the proline ring onto the S1′ pocket for this novel generation of FD inhibitors (**structures 3**). Separately, fragment-based screening (FBS) yielded molecule **4a** as a very weak FD ligand (NMR K_d_ = 1600 μM). The synthesis of the water-soluble analog 4**b** enabled the solving of the co-crystal structure with the protein. The aromatic (planar) indole moiety was squeezed between Lys_192_ and the self-inhibitory loop, inside the narrow S1 pocket, the primary amide group was engaged in H-bonding interactions with the side chain of Arg_218_ and the backbone carbonyl oxygen of Thr_214_, and the remaining benzyl ether moiety was exposed to solvent space. Molecule **5**, a hybrid structure of molecules **3C** and **4**, turned out to be a moderately potent (IC_50_ = 0.50 μM in the esterolytic assay) and selective FD inhibitor [39].

Additional medicinal chemistry work aimed to not only improve potency but also optimize other parameters, such as cLogP, binding efficiency index (BEI), aqueous solubility, permeability with no efflux potential, human plasma protein binding, HERG and CYP450 activities, and human clearance. The strategy was to work on one structure segment at a time (Figure 4), so that they achieve efficient optimal binding in each one of the three active site hot spots, S1′, S2′, and S1, respectively. The central (S)-proline scaffold was absolutely essential for activity contributing to the required pharmacophore bent shape. The incorporation of either a fused *trans*-4,5-cyclopropyl ring or a fluorine atom of the *R* configuration on the proline ring improved potency significantly by taking advantage of supplemental ligand interactions within an indentation formed because of the outward His_57_ conformation (Figure 5). It is also known that stereospecific substitution at the C-4 proline carbon with fluorine or other groups (-OH, -NH_2_) favors overwhelmingly one of the two possible proline ring puckerings (*exo*- over *endo*-) and enhances the conformational preference of the proline N-C(O) amide bond (*trans*- over *cis*-) [40]. Moving to the P2′segment, substituents (-OCF_3_, -CF_3_) at the *meta*-position of the aniline moiety, capable of engaging in hydrophobic interactions within the S2′ pocket were optimal. Replacing those groups with the larger halogens (Br > Cl) maintained potency. Furthermore, introducing fluorine at the *ortho*-position or having the pyridine ring shown as a benzene ring surrogate increased potency further, due to the resulting elevated acidity of the adjacent amide hydrogen interacting with the backbone carbonyl oxygen of Leu_40_. Choosing the most appropriate P1-P1′ spacer and P1 aromatic moiety were the last key steps of this process. Replacing the urea NH of molecule **5** with a -CH_2_- provided the optimal vector for the entrance of the P1 moiety into the S1 pocket (Figure 5). Finally, repositioning the indole nitrogen of **5** and utilizing an indazole ring system instead substituted with either the carbamoyl group or the acetyl group provided the optimal P1 core structure.

The extensive lead optimization process summarized herein identified molecule **6** (Figure 5a) as a highly potent FD inhibitor with IC_50_ values of 0.006 μM and 0.05 μM in the esterolytic assay and the MAC complex formation assay, respectively. It was also highly selective for FD over a large group of other serine proteases.

The previously mentioned H-bond interactions with Gly_193_ of the oxyanion hole and Leu_40_ anchor the inhibitor onto the S1′ active site pocket (Figure 5b). The axial methylene group of the fused cyclopropyl ring contributes further to S1′ binding via hydrophobic interactions with the surrounding amino acids (Figure 5c). The plane of the indazole moiety is perpendicular to that of the neighboring spacer, a critical feature of the required architecture for efficient binding to the narrow and deep S1 pocket having both hydrophobic and hydrophilic characteristics (Figure 5c). The engagement of the primary amide substituent on the indazole ring in an extensive network of H-bonds within this pocket contributes significantly to the overall ligand affinity (Figure 5b) [41]. The coupling of a bromo-substituted pyridyl amine to the proline amino acid extends the molecule towards the hydrophobic S2′ hot spot, providing the handle for an additional critical interaction, namely the interesting halogen bond with the Trp_141_ backbone carbonyl oxygen (Figure 5b). The pyridine nitrogen increases the strength of this not-so-common bonding interaction [42].

**Figure 5 ijms-23-07216-f005:**
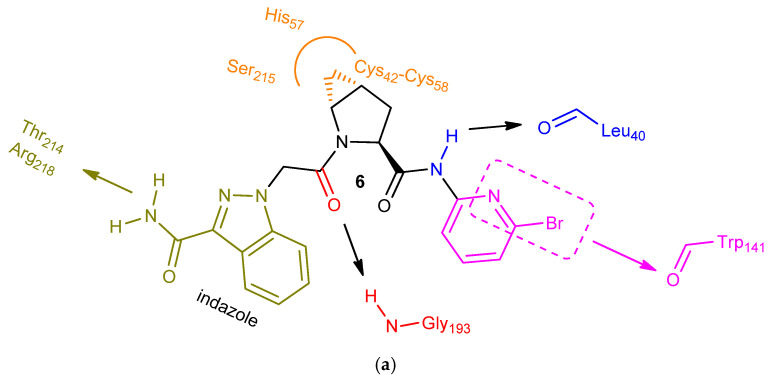
(**a**): Simplified view of the structural features of **6** and the respective key ligand–protein interactions responsible for both potency and selectivity. (**b**,**c**): Optimized analog (**6**) of the (S)-proline series binds to the U-shaped S1-S1′-S2′ active site cleft of the latent FD conformation. The in silico model shown here was generated in the laboratory of K. A. Agrios (Villanova University, Villanova, PA, USA) using Glide [43].

Molecule **7** (Figure 6) was a secondary lead structure of the (*S*)-proline series of small molecules described herein. It was an effective FD specific ligand based on surface plasmon resonance experiments (K_d_ = 0.006 μM, k_on_ = 1.64 × 10^6^ M^−1^s^−1^). It did not have any considerable activity (<10 μM) against a panel of other proteases, kinases, ion channels, and other receptors. In vitro and in vivo experiments in human FD knock-in mice demonstrated the ability of **7** to block AP function by exclusive FD inhibition without affecting CP and LP activities. Moreover, compound **7** showed in vivo efficacy in a human FD KI mouse AMD model. However, the overall profile of molecule **6** was more favorable than that of analog **7** meaning better Caco-2 cell permeability, no efflux, and better oral bioavailability (100% versus 28%, respectively) in the rat. Molecule **9** was another interesting hit of their FBS campaign offering a distinct binding motif to the S1′-S2′ region of the active site. The carboxylate carbonyl of **9** (shown in red) was engaged in H-bonding with Gly_193_ of the oxyanion hole, and the *trans* configured cyclopropyl scaffold directed the adjacent phenyl substituent (shown in purple) towards a π–cation interaction with the guanidinium group of Arg_151_. Merging the latter fragment with the S1-S1′ binding core of molecule **6** yielded a new chemotype of latent FD inhibitors represented by molecule **8**.

**-** 
**“Me too” latent FD inhibitors**


FD inhibitor **6** did not advance into clinical development even though the publicly shared pre-clinical profile was reasonable. However, the elegant work by the Novartis team had ignited the interest of the drug discovery community. Scientists at Achillion aimed to develop their proprietary small molecule FD inhibitors (Figure 7). Their strategy relied on functionalizing further the indazole ring of molecule **6** while leaving intact its remaining core structure. They were able to establish their own intellectual property (IP) position by connecting a plethora of chemical moieties to specific carbon atoms of the indazole ring. Their medicinal chemistry efforts culminated in the identification of two molecules, designated later as danicopan and vemircopan, currently undergoing human clinical trials (discussed later in this review). Danicopan has the (*R*)-4-fluoroproline ring as the P1′ residue taking advantage of an interaction between the C-F and the His_57_ backbone carbonyl dipoles [39,44]. The proline ring of vemircopan is decorated with the *trans*-4,5-cyclopropyl moiety, as it was the case with **6**, albeit having C-4 fully substituted with an additional methyl group. Furthermore, the introduction of a methyl substituent on the pyridine ring may offer an extra conformational bias to better position the ring inside the S2′ pocket. We speculate that the “follow on” vemircopan, a slightly modified version of danicopan, attempts to address pharmacokinetic issues associated with the latter. Both molecules are approximately equally potent as **6** in a wide spectrum of functional in vitro assays, modeling both prototype diseases of complement activation: PNH and complement mediated TMAs [45]. An issued US patent (**10,092,584**, Achillion) claims (among others) analogs **10** and **11**. It is possible that a third clinical candidate has been selected based on these lead structures.

#### 2.2.2. First-Generation of Inhibitors Targeting the “Unlocked” FD Conformation

The Novartis team also tried to identify chemotypes of reversible FD inhibitors that lack peptidomimetic structural features ((*S*)-proline series) and have a differentiated ADMET profile [46]. The weak ligand **4b** (Figure 8), binding only to the S1 pocket and discussed earlier in this review served as their starting point. Their strategy relied on utilizing the essential indole core of **4b** and trying to augment the structure via C-4 towards the S1′ pocket. They designed molecule **12**, where the amide linker was supposed to be engaged in an enzyme-bound water-mediated H-bonding interaction with Gly_193_ and Ser_195_ as well as in a direct H-bonding interaction with Ser_217_. They envisioned gaining potency from an ionic interaction with the flexible acidic 60-loop on top of the S1′ pocket by installing a basic *meta*-benzylamine moiety at C-4. The slightly increased affinity of ligand **12** (K_d_ = 1000 μM) was accompanied by a huge surprise after solving the structure of the crystal complex. The binding pose of **12** was flipped by 180° (**12**) with the basic benzylamine moiety forming a salt bridge with Asp_189_, thereby disrupting the respective interaction with Arg_218_. As a result, the self-inhibitory loop was displaced due to significant conformational changes and the guanidine of Arg_218_ ended up being exposed to solvent. It is important to note that the S1 pocket still remains narrow and His_57_ still adopts an outward (non-catalytic) conformation. The reader of this review should relate these findings with the excellent crystallography work describing FD switching to an S1 unlocked conformation after binding onto the FBC3b complex [29].

They envisioned earning an added benefit in terms of increased selectivity over the panel of other human serine proteases by targeting this unique FD conformation. The replacement of the reverse amide of **12** with a phenyl moiety furnished molecule **15** (Figure 9) as an improved, albeit still weak, FD inhibitor (K_d_ = 20 μM (NMR) and IC_50_ = 54 μM in the thioesterolysis assay). Modeling as well as crystallography work confirmed the positioning of the tetrahydronaphthalene moiety in the S1β pocket, while the amide NH is close to the Arg_218_ carbonyl. The observation of a single succinic acid molecule coming from the crystallization buffer, in the vicinity of the oxyanion hole as well as the fact that His_57_ had adopted the canonical (catalytic) conformation for the first time, were quite interesting too. Replacing the bulky hydrophobic moiety with phenyl acetic acid (**15** to **16**) was supposed to provide enhanced binding interactions with the basic side chain of Arg_218_ and therefore increase potency [47]. That actually happened (IC_50_ of **16** was 120 nM in the thioesterolysis assay) with the caveat that the carboxylic acid group was found instead to H-bond with His_57_, Gly_193_, and Ser_195_. Of note is that the co-crystal structures of **15** and **16** show the overlapping of the succinic acid with the -COOH of **16**. Even though installing the phenyl acetic acid moiety proved to be instrumental in increasing potency, selectivity was somewhat compromised. Molecule **16** was a low μM inhibitor of both Factor XIa and tryptase–β2. It also exhibited poor permeability and hence low oral bioavailability in mice (10%). The ensuing replacement of the polar amide linker of **16** with a specific ether linker (**17**) and the addition of a hydroxymethyl substituent on the benzylic amine carbon (**18**) improved potency, permeability, and selectivity (640, 540, and 2500–fold window for Factor XIa, tryptase–β2, and urokinase, respectively).

**-** 
**“Me too” inhibitors targeting the “unlocked” FD conformation**


Another pharmaceutical company (Biocryst) utilized FD inhibitor **18** as their lead structure. Their efforts culminated in a patent publication (**US2019/0345135**) claiming a plethora of analogs, closely mimicking structure **18**. Careful examination of the patent claims and experimental data point to **19** (Figure 10) as their main lead. Installing a benzofuran core as a replacement for the central phenyl moiety of **18** was their key intellectual property (IP) strategy. Adding substituents on the arrow pointed carbon atoms of **19** and using other heterocyclic rings, besides the furan ring, yielded potent analogs too (Figure 10). We would also like to highlight claimed and synthesized compounds **20**, **21**, and **22** aiming to reach beyond the oxyanion hole and towards the S2′ protein hot spot.

#### 2.2.3. Second-Generation of Latent FD Inhibitors

Another patent publication (**WO2017/136395**, Biocryst) claims a distinctive chemotype (structure **23**, Figure 11) of small molecule FD inhibitors mimicking FD inhibitors **6** and **7**. Based on the patent claims and experimental data, suitably N-substituted glycine can serve as a viable replacement of the (*S*)-proline scaffold impeded in molecules **6** and **7**. Biocryst has advanced small molecule **BCX9930** (the structure has not been disclosed to the public) into phase 1 and phase 2 clinical studies. We assume that the structure of BCX9930 is either identical or a close analog to one of the two core structures (**19** or **23**) claimed in the respective patent applications.

## 3. Clinical Development of Complement Inhibitors

### 3.1. C5 Inhibitors in PNH

Eculizumab and ravulizumab are monoclonal antibodies aiming at C5 inhibition and are considered state-of-the-art therapy for PNH patients [48]. They both prevent C5 division into C5a and C5b, with ravulizumab offering long-term action and therefore an easier administration scheme. Additional C5 inhibitors, such as crovalimab, are also under advanced clinical development, with the advantage of easy subcutaneous administration [49]. Although C5 inhibitors block the complement pathway and cease the subsequent complement-mediated hemolysis, they do not attend to the extravascular hemolysis-induced anemia [50], with several clinical manifestations. Indeed, the research aims to relieve patients of the remaining anemia during anti-C5 therapy. It has been recently demonstrated that only 60% of PNH patients on eculizumab reached and maintained hemoglobin normalization [51].

### 3.2. Development and Application of C3 Inhibitors and Similar Molecules

First described, preclinical C3-mediated extravascular hemolysis paved the way in the treatment of PNH, addressing the proximal complement inhibition [7]. Clearly, interruption at the C3 stage (or upscale) of the alternative pathway blocks cells’ opsonization with C3 fragments [52] and eventually brings extravascular hemolysis to a screeching halt [53]. Clinical data have also proven the superiority of C3 inhibition. The randomized PEGASUS trial has shown the superiority of pegcetacoplan as a C3 inhibitor administered subcutaneously compared to eculizumab in a large population of PNH patients [11].

**-** 
**Factor D inhibitors**


Danicopan, an oral blocker of factor D, selectively and effectively inhibits proximally the alternative complement pathway, as shown in open-label single-arm phase 2 studies [54,55]. In the first one, eight out of ten PNH patients completed the study, with orally given treatment (100–200 mg three times daily) as a monotherapy, having received no previous treatment for PNH. Danicopan significantly lowered LDH levels (*p* < 0.001) and meaningfully improved hemoglobin levels (1.1 and 1.7 g/dL rise from baseline at 4 weeks and 12 weeks, respectively, *p* < 0.005). Adverse events included headache, nasopharyngitis, and cough. All in all, danicopan hinders intravascular hemolysis, completely restricts C3-mediated extravascular hemolysis, and results in substantially elevated hemoglobin levels [54].

Poor responders to eculizumab comprised the investigated group in the second phase 2 study [55]. Danicopan was administered orally as an add-on treatment (100–200 mg three times per day) to a group of 12 patients who continued to have high transfusion rates, despite adequate eculizumab therapy. Eleven out of twelve patients were evaluated at week 24, showing a mean hemoglobin rise of 2.4 g/dL. Most importantly, during the studied period, only one patient required one transfusion of 2 units, in comparison to a total of 34 transfusions or 58 units administered to ten patients in the preceding 12 weeks. Overall, danicopan showed high efficacy rates in PNH patients not responding to eculizumab, in terms of reduction of extravascular hemolysis and improvement in hemoglobin levels. Moreover, danicopan improved transfusion rates, bilirubin ranges, reticulocyte levels, and patient-reported fatigue (as evaluated with Mean Functional Assessment of Chronic Illness Therapy (FACIT)–Fatigue score).

Further trials, which will consolidate the treatment of PNH patients with this FD complement blocker, are ongoing. Danicopan is being studied in a phase 3 study as an add-on therapy to a C5 inhibitor (eculizumab or ravulizumab) in patients with paroxysmal nocturnal hemoglobinuria, who have clinically evident extravascular hemolysis (NCT04469465). Moreover, a next-generation inhibitor of factor D, a danicopan analog (ACH-5228 or ALXN-2050) is being analyzed as monotherapy (NCT04170023).

## 4. Conclusions and Future Perspectives

The miracle of complement inhibitors as “orphan drugs” has dramatically improved morbidity and mortality in patients with otherwise life-threatening complementopathies. Efficacy has been significantly improved by upstream inhibition in patients with PNH. Further studies remain to show the safety and efficacy of upstream inhibition in other complementopathies. In addition, cost and availability issues are major drawbacks of current treatments. Therefore, further studies and developments are warranted to address the unmet clinical needs in the field of complementopathies. Orally administered small molecule complement therapeutics could become major drivers moving forward. Avacopan (Tanveos^R^) is currently undergoing clinical trials in additional indications. The medical community and the patients suffering from these difficult-to-treat and rare diseases are awaiting the phase 2 and phase 3 clinical results of the three FD inhibitors (danicopan, vemircopan, and BCX9930) as well as the FB inhibitor (iptacopan). The potential of any of these small molecules to show clinical efficacy as monotherapy represents the next big hurdle of this process. The outcome of the battle between FD and FB inhibition is also anticipated with great interest. Currently available chemotypes of FD inhibitors presented in this review will provide the blueprint to design new ones with superior efficacy and ADME profiles.

## Figures and Tables

**Figure 1 ijms-23-07216-f001:**
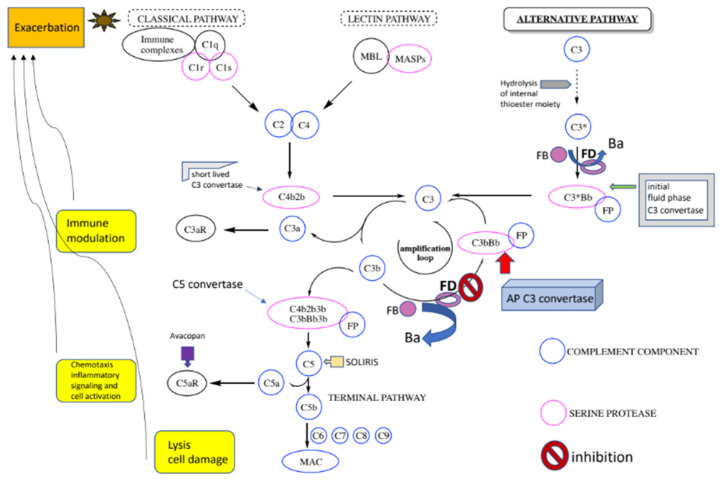
**The complement system with a focus on the AP**: A spontaneous hydrolysis of a thioester bond within C3 (C3 “tickover”) results in a conformationally altered C3 (designated C3 (H_2_O) or C3*) that makes the recruitment of FB possible. FB undergoes its own conformational change and is cleaved by another serine protease, factor D (FD), to generate fragments Ba and Bb. The latter contains the serine protease domain of FB and remains bound to the complex designated C3*Bb and the initial fluid phase AP C3 convertase. This convertase catalyzes the cleavage of C3 molecules to generate fragments C3a (anaphylatoxin) and C3b (opsonin). The same C3 proteolytic cleavage can also be catalyzed by the CP/LP C3 convertases, C4b2b. The released C3b components can enter into the critical amplification loop of the AP where complexes of foreign surface bound C3b and FB are similarly cleaved by FD to generate the predominant AP C3 convertases, C3bBb, on the targeted surfaces capable of additional C3 proteolytic cleavage. The C3b components also have the option of binding to either of the C3 convertases to form the respective C5 convertase complexes responsible for the breakdown of C5 molecules to release C5a and C5b. The latter interacts with C6, C7, C8, and C9 to form the membrane attack complex (MAC), which punctures the surface of some pathogens, resulting in subsequent cell lysis.

**Figure 2 ijms-23-07216-f002:**
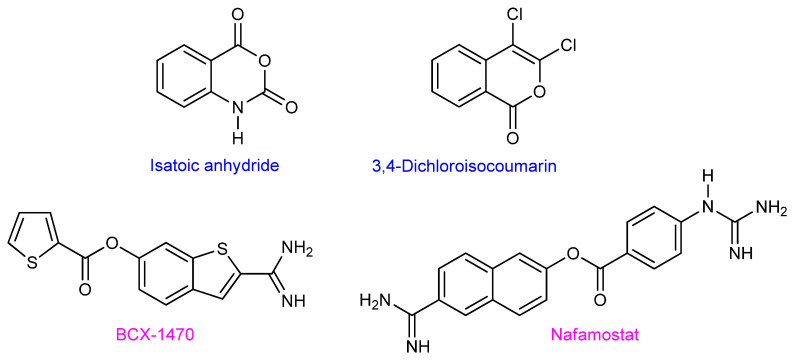
Structures of reversible and non-selective FD inhibitors. Isatoic anhydride, 3,4-Dichloroisocoumarin, and BCX-1470 were used primarily as molecular probes. Nafamostat has found some limited pharmacological use (IV administration) for patients with acute kidney injury and pancreatitis.

**Figure 3 ijms-23-07216-f003:**
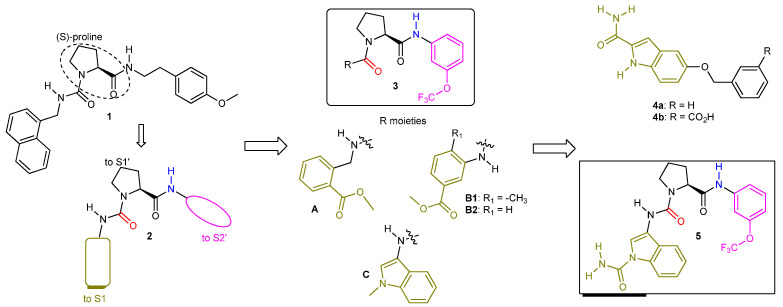
**Evolution of KLK7 inhibitor 1 to FD selective inhibitor 5**. The solved co-crystal structure of molecule **3A** with human FD revealed a bent conformation for **3A** fitting into the U-shaped S1-S1′-S2′ FD cleft, analogous to the **1**-KLK7 pair. The *ortho*-positioned ester moiety of molecule **3A** acted as a hydrogen bond acceptor interacting with the Arg_218_ side chain while the -OCF_3_ group, being perpendicular relative to the benzene ring plane, was engaged in a hydrophobic interaction. Moving the ester moiety at the meta-position (molecules **3B1** and **3B2**) required the concomitant presence of a small alkyl group R_1_ at the *ortho*-position for FD inhibitory activity (IC_50_ of **3B1** was 17 μM while **3B2** was inactive). The *ortho*-methyl group enforces a properly distorted (non-planar) geometry around the phenyl ring, facilitating its entrance into the narrow S1 pocket. Molecule **3C** (IC_50_ = 20 μM) has an indole ring as the S1 binding moiety where the -NCH_3_ is found in close proximity to the Arg_218_ side chain and to overlap with the indole carbon of molecules **4** carrying a primary amide group.

**Figure 4 ijms-23-07216-f004:**
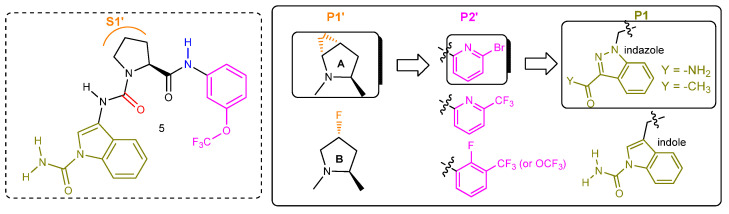
Lead optimization of **5** by segment (P1′, P2′, P1) and in the order shown.

**Figure 6 ijms-23-07216-f006:**
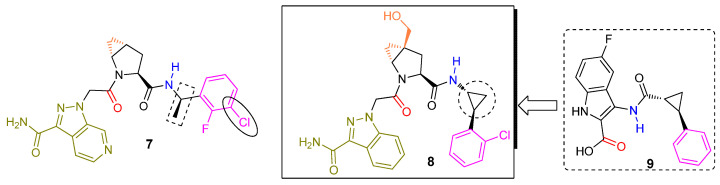
Additional lead structures of (S)-proline-based latent FD inhibitors. Molecule **8** was included in a published patent publication (**WO2014/002051**, Novartis). It was the most potent inhibitor among 87 analogs synthesized and tested in the human FD thioesterolytic assay (IC_50_ = 0.022 μM).

**Figure 7 ijms-23-07216-f007:**
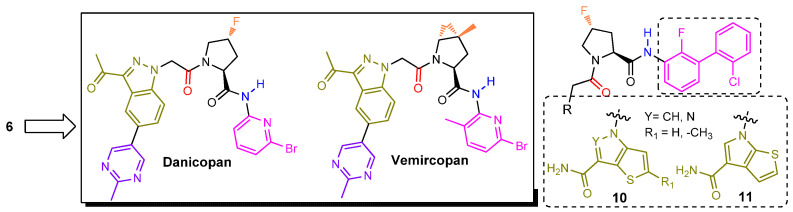
Both danicopan and vemircopan are very close replicas of molecule **6**, having a methyl-substituted pyrimidine ring attached to one of the indazole carbon atoms. Our own modeling studies of danicopan (not included in this review) revealed that the added pyrimidine ring is oriented towards solvent space and therefore should have no impact on its binding affinity.

**Figure 8 ijms-23-07216-f008:**
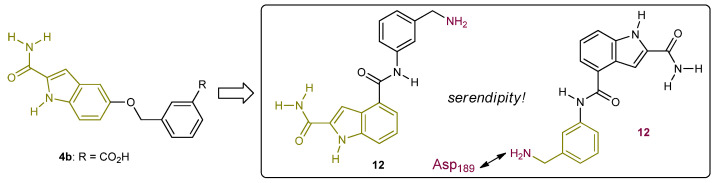
Serendipitous discovery of FD ligand **12** binding to the enzyme unlocked conformation (S1 open conformation).

**Figure 9 ijms-23-07216-f009:**
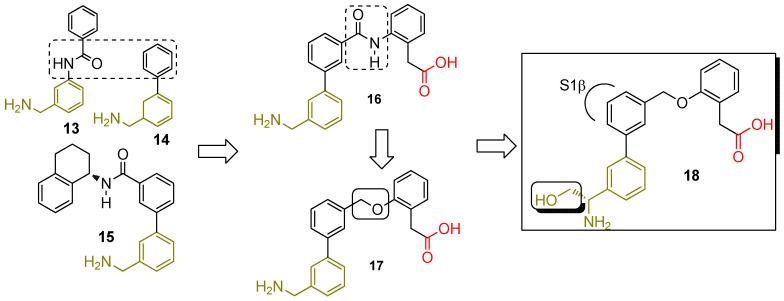
Structure-based drug design led to the discovery of potent and selective FD inhibitor **18** targeting the “unlocked” FD conformation. **18** had IC_50_ values of 12 nM and 260 nM in the thioesterolysis and human 50% whole blood MAC assays, respectively, accompanied by 83% oral bioavailability in mice.

**Figure 10 ijms-23-07216-f010:**
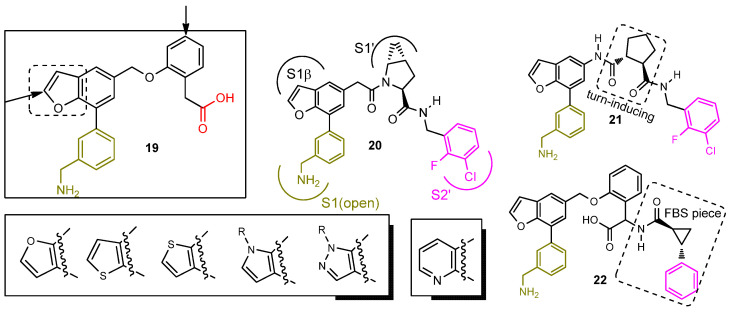
Benzofuran analogs (**US2019/0345135**, Biocryst) of potent and selective FD inhibitor **18**. Analogs **20** and **21** appear to be hybrid structures of potent FD inhibitors **7** and **19,** while **22** is built based on inhibitors **19** and **9**. All analogs shown are reported to inhibit FD activity in the respective thioesterolysis assay with an IC_50_ < 1 μM.

**Figure 11 ijms-23-07216-f011:**
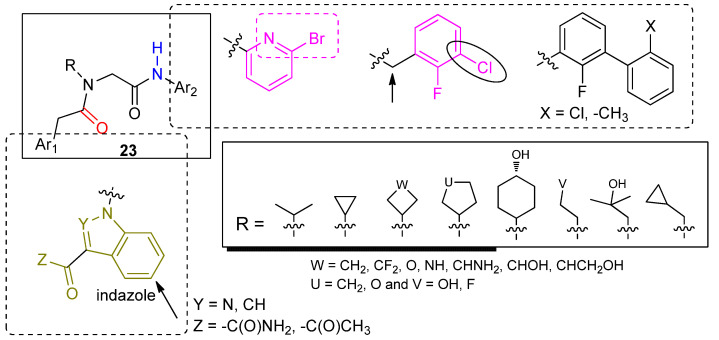
N-Substituted glycine-based FD inhibitors with affinity for the locked (latent) enzyme conformation. The substituents R shown appear to be optimal for potency (IC_50_ < 1 μM). The glycine scaffold is tethered to an appropriately substituted indazole or indole ring system to ensure binding to the S1 (locked) pocket. The amines bonded to the glycine carbonyl (upper right box) provide the necessary engaging elements for interactions within the S2′ hot spot.

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
