# Peer review of "Novel Insights into Factor D Inhibition"

_ijms, 2022, doi:10.3390/ijms23137216_

Round 1

Reviewer 1 Report

This is a very detailed review of the development of FD inhibitors. It is focused on the structural biologist and medicinal chemist--I am not either. Therefore, I found it very difficult to read, with what I would consider far too much detail on the medicinal chemistry for a journal such as IJMS. However, the authors are knowledgeable and the content is important and timely. 

I would like to see this article rewritten and rebalanced, with approximately (for example) 25% of the article describing factor D and its role in human disease, deficiency, etc, 30% of the article describing complementopathies in greater detail, and expanding beyond only PNH and hematologic disorders (e.g. C3 glomerulopathy, opthalmic disorders, etc), 10% of the article describing the development of FD inhibitors (currently about 90%) and the rest of the article providing detailed descriptions of FD clinical trials and future directions. 

Minor points include the fact that the figure legends need to be greatly expanded (perhaps allowing the manuscript text to be shortened) and the labels within the figures need to be larger so they are legible. In some cases the text refers to a number and it is not clear whether this is the figure number or the compound number. This needs significant editing.

Greater use of molecular models of FD-inhibitor interactions in all of the figures would also be helpful.  

Author Response

We thank the Reviewer for the kind comments.

The novelty added by this Review in current literature is the explanation of factor D inhibitors' drug discovery. Prompted by the Reviewers' suggestions however, we have extensively revised our manuscript to include more clinical data and additional complementopathies. We have also deleted many details of factor D inhibitors.

We have also edited the Figure legends.

Finally, the two models included in this review were generated by one of us and that is why we have the legal rights to include them in this review.  Several models can be found in the following publications (references 24, 40, 42, 47, 48). However we do not have permission by the respective publishers to include those in our review.

Reviewer 2 Report

Eleni Gavriilaki, M.D., Ph.D., with colleagues, presents a review paper describing recent advances in Factor D inhibition. The discussed area is for sure an emerging field of medicine, thus, every paper contributing to the field is of great importance. I went carefully through the whole draft and I do not feel that the paper is publishable in its current form, albeit, after a deeply conducted review it might be reconsidered for publication. Please follow my majors: 

Firstly, I must recommend extensive English language editing. The draft contains a bunch of wrongly used words as well as terms. There are lots of minor and major grammatical errors that make the whole draft really hard to follow. Please avoid mixing American and British English. The native speaker should go over the draft and fix all the issues present there. 

Secondly, I am missing the correct structure for the draft. The first part of the paper definitely should contain more biological/molecular characteristics of Factor D. Now, the introduction part is very hard to get along with for the people who are not an expert in this field. Since the Authors discuss very specific and narrow areas they should provide definitely more basic information regarding Factor D for the readers. I am also missing the part about the differences and comparison between 1) the lectin pathway, 2) the classical pathway, and 3) the alternative pathway of complement (APC) which converges on the common central component C3.

The paper is very chemical-focused. I definitely would recommend introduction more explanatory form of the discussion - the Authors aim to publish a paper in a biological journal rather than chemistry, so the audience needs also some very basic scientific information e.g. how the introduction of the additional ring to the structure impacts basic pharmacodynamic/kinetics properties and half-life of the compounds etc. Please try to guide the readers through the paper. Not only give very technical and "hard to digest" chemical data. 

Moving further, there is no information given regarding the following PRISMA guidelines and description of the searching strategy, which is so crucial for review papers. The Author should include at least: Data sources and searches, Study eligibility criteria, Study selection process, Data extraction, and study quality assessment (assessing the risk of bias (ROB) for each included study), Data synthesis. MeSH terms and keywords are necessary to be included. For each step, it is necessary to explain to the reader with pictures or tables. It is necessary to explain what was drawn at each step to lead to the result. Moreover, a figure showing the PRISMA-based workflow should be drawn. After that, a discussion is valuable. 

Since the structure of the chemical compound is a key player in the paper additional supplementary files showing 3D structure could be included. 

Please expand the figure legends - they should be readable and informative when standing alone.

I am looking forward to the revised version of the paper. 

Author Response

We thank the Reviewer for giving us the opportunity to improve our manuscript. We have made a great effort to implement the Reviewer's suggestions into the revised manuscript. We submit the revised manuscript with track changes in the parts of major changes so that the manuscript is readable.

  1. The native speaker (Konstantinos Agrios) has revised the draft in order to harmonize the language.
  2. We have emphasized (in the revised version) the differences between the classic, lectin and alternative pathways
  3. We understand the Reviewer's concern and therefore, we eliminated this type of comments in the revised version. We agree that these comments are not suitable here and they do not add extra value to our review. Moreover, elaborating further on these comments would increase the length of the text significantly.
  4. The present manuscript is a narrative review and not a systematic one. We have added this term in both the abstract and the aim. Therefore, PRISMA workflow is not applicable.
  5. Figure legends have been revised in order to be readable and informative

Reviewer 3 Report

Gavriilaki et al. present a very nice and comprehensive review manuscript on the factor D inhibition. There is only one deficiency that authors might need to consider to improve, which would result in a better readability of their manuscript. All figure legends must be more descriptive. For example Figure 3 demonstrates a nice historical perspective on factor D inhibitors, however, a further description would provide better insight into authors' intended message. 

Author Response

We are grateful to the Reviewer for recognizing the strengths of our manuscript. We have edited all Figure legends as requested.

Round 2

Reviewer 1 Report

This manuscript has had some significant additions providing some background on factor D. I had been asking for some condensation and streamlining of the medicinal chemistry part of the manuscript so it would be of greater interest and understanding to biological scientists and clinicians rather than medicinal chemists. This is lacking. However, the manuscript is a nice addition to the literature. 

Author Response

We thank the Reviewer for the kind comments. Based on both Reviewers' comments, we have made some refinements in the required fields.

Reviewer 2 Report

The paper has been improved, albeit, majors are still present.

Firstly, even for narrative review, there are some parts of PRISMA that must be included, namely: please provide the exact keywords that have been used for literature searching, please list the databases that have been included in the literature screening as well as all the clinical trials that have been included.

The Authors did not include a comment nor change anything regarding the comment: The paper is very chemical-focused. I definitely would recommend introduction more explanatory form of the discussion - the Authors aim to publish a paper in a biological journal rather than chemistry, so the audience needs also some very basic scientific information e.g. how the introduction of the additional ring to the structure impacts basic pharmacodynamic/kinetics properties and half-life of the compounds etc. Please try to guide the readers through the paper. Not only give very technical and "hard to digest" chemical data. 

The Authors did not appropriately address the comment: I am missing the correct structure for the draft. The first part of the paper definitely should contain more biological/molecular characteristics of Factor D. Now, the introduction part is very hard to get along with for the people who are not an expert in this field. 

The Authors did not reply to the comment: Since the structure of the chemical compound is a key player in the paper additional supplementary files showing 3D structure could be included. 

The figures' legends are still not readable when standing alone. 

Author Response

-"The Authors did not include a comment nor change anything regarding the comment: The paper is very chemical-focused. I definitely would recommend introduction more explanatory form of the discussion - the Authors aim to publish a paper in a biological journal rather than chemistry, so the audience needs also some very basic scientific information e.g. how the introduction of the additional ring to the structure impacts basic pharmacodynamic/kinetics properties and half-life of the compounds etc. Please try to guide the readers through the paper. Not only give very technical and "hard to digest" chemical data."

We thank the Reviewer for the thorough reading of the manuscript. Specific comments have been deleted from the revised manuscript.

In general, we all know that drug discovery and development is a multidimensional field. It involves medicinal chemistry, molecular biology, biochemistry, cell biology, pharmacology, pharmaceutics and in the end clinical studies. In our opinion, experts and practitioners in each one of these fields should find this review paper useful. In this review paper, we aim to present a concise and comprehensive view of the progress that has been made towards the development of orally active FD inhibitors for the treatment of complementopathies. We present the path followed by the medicinal chemists to discover small molecules suitable for that purpose. We discuss the special structural and functional characteristics of the protein (FD). Then we present the SAR (Structure Activity Relationships) profiles of all small molecule FD inhibitors that have appeared in the literature (including the patent literature) up to now and how they affect biochemical potency, selectivity as well as other parameters.  The medicinal chemistry section, which is larger in size than the clinical section, could be very useful to those who will attempt to discover the next generations of small molecule FD inhibitors. Clinicians will find more useful the clinical section of this paper. More importantly, the correlation of the two will be useful for all readers. The structure of a molecule undergoing clinical trials is absolutely critical for the success or failure of that molecule in the clinic. For instance, the phase 2 clinical trial of BCX9930 was recently halted due to the fact that many patients receiving the 400 mg dose of this drug experienced high levels of creatinine. The respective clinical trial of Danicopan did not encounter similar issues. It is very safe to assume that the specific differentiation of these two molecules in the clinical setting can be attributed to their respective molecular structures. We believe that all readers of this review paper will find very useful to be able to correlate molecular structures, SAR profiles and clinical findings. Closing, we believe that the style of presentation and writing in section 3 (medicinal chemistry, biochemistry) is similar to that in section 4 (clinical results). Section 4 presented clinical data for one molecule (mainly) without offering any explanations for terms used and data presented.

"The Authors did not reply to the comment: Since the structure of the chemical compound is a key player in the paper additional supplementary files showing 3D structure could be included. "

I can only assume (the reviewer dos not specify with a name or a number) that the comment “chemical compound” refers to the molecule discovered and owned by Achillion, namely Danicopan. There are no in silico molecular models of Danicopan in the literature up to May 2022. In fact reference 46 introducing Danicopan and its profile does not even disclose the chemical structure of Danicopan. One of the authors (Konstantinos Agrios) has worked extensively on small molecule FD inhibitors and owns several molecules that are novel potent and selective FD inhibitors. He has not published yet any of his research findings in order to protect his IP position. In the context of his research work, he has generated an in silico model of Danicopan and he plans to include it in one of his own future publications. Of note, this is a review paper intended to present data that are already published, and not original research, not published yet. To validate further our statement, the two in silico models of molecule 6, included in this review, were generated by Konstantinos Agrios. An in silico model of this exact molecule (molecule 6) has already been included by the Novartis scientists in one of their publications. Closing, in silico models of other known FD inhibitors can be found by future readers of this paper in the following publications (references 7, 38, 40, 42, 47 and 48). We do not have permission by the respective publishers to copy and include models from those papers in this review paper.

"The figures' legends are still not readable when standing alone."

Looking at all figures we thought that it is necessary to edit Figure 8 so that we could increase the size of the atom labels in all structures. The text underneath each figure can be modified in terms of size based on the technical guidance offered by the publisher of this journal. We will definitely follow their guidelines.

Round 3

Reviewer 2 Report

The Authors addressed, at least partially, my majors as well as most of my minors. 

If the Editor feels that the paper fits the journal scope in terms of balance between chemical and biological weights, I am happy to recommend the acceptance as it is. 

Please only fix some template issues that are present in the current version. 

Best.